# StyleBART: Decorate Pretrained Model with Style Adapters for Unsupervised Stylistic Headline Generation

**Hanqing Wang**[1*]    **Yajing Luo**[1*]    **Boya Xiong**[1]    **Guanhua Chen**[2]    **Yun Chen**[1†]

[1]Shanghai University of Finance and Economics
[2]Southern University of Science and Technology
{whq, luoyajing, xiongboya}@163.sufe.edu.cn
chengh3@sustech.edu.cn  yunchen@sufe.edu.cn

## Abstract

Stylistic headline generation is the task of generate a headline that not only summarizes the content of a news, but also reflects a desired style that attracts users. As style-specific news-headline pairs are scarce, previous research has focused on unsupervised approaches using a standard headline generation dataset and mono-style corpora. In this work, we follow this line and propose StyleBART, an unsupervised approach for stylistic headline generation. Our method decorates the pretrained BART model with adapters that are responsible for different styles and allows the generation of headlines with diverse styles by simply switching the adapters. Different from previous works, StyleBART separates the task of style learning and headline generation, making it possible to freely combine the base model and the style adapters during inference. We further propose an inverse paraphrasing task to enhance the style adapters. Extensive automatic and human evaluations show that StyleBART achieves new state-of-the-art performance in the unsupervised stylistic headline generation task, producing high-quality headlines with the desired style. Code is available at https://github.com/sufenlp/StyleBART.

## 1  Introduction

The sequence-to-sequence-based neural headline generation (HG) model (Cao et al., 2018; Lin et al., 2018; Song et al., 2019) has demonstrated its ability to generate factual, concise, and fluent headlines (Chopra et al., 2016). Yet, the headlines are also expected to have stylistic attributes to draw more attention of the audiences. To address the problem, Jin et al. (2020) propose the task of Stylistic Headline Generation (SHG), which aims to generate a headline with a specified style such as humorous, romantic and clickbaity. However, acquir-

*Co-first author with equal contribution
†Corresponding author

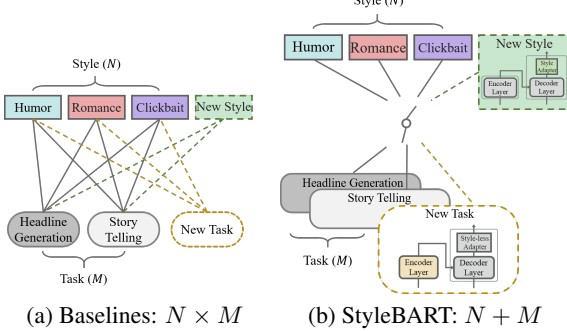

(a) Baselines: $N \times M$    (b) StyleBART: $N + M$

Figure 1: Our StyleBART is training-efficient. (1) For a combination of $M$ tasks and $N$ styles, StyleBART only trains $M$ task-specific models and $N$ style-specific modules, then combines them at inference time to support each stylistic generation task. Previous works train separate $M \times N$ models. (2) For a new style (task), StyleBART only trains a new style module (task model), however, baselines have to train many additional models to combine the new style (task) with all existing tasks (styles).

ing enough parallel data for SHG is almost impossible, as the creation of headlines with specific styles demands creativity and often consumes significant effort. Hence, researchers (Jin et al., 2020; Zhan et al., 2022) in turn explore unsupervised approaches which only require a standard headline generation dataset and non-parallel stylistic text corpora.

Some existing works propose a pipeline method (Sudhakar et al., 2019; Dai et al., 2019; Krishna et al., 2020) which firstly generates a plain headline and then introduces the pre-specified style with a style transfer model. However, this pipeline method require two models during inference, which brings additional latency and storage cost. Some other works (Jin et al., 2020; Zhan et al., 2022) jointly learn a plain headline generation model on news-headline pairs, and a denoising autoencoder on the stylistic text corpus. However, these approaches require to carefully design the scheduling

and parameter sharing mechanisms between tasks. Moreover, as the task of headline generation and style learning are entangled, the entire model has to be rebuilt when facing a new style or task.

In this paper, we propose StyleBART, an unsupervised SHG model based on the BART model (Lewis et al., 2020). Instead of using a multi-task learning framework, StyleBART disentangles the style and the specific task (e.g. headline generation) by the design of model architecture and training strategy. Intuitively, this enables Style-BART to separately train the style and task modules, then combine them as required during inference (Figure 1). Hence, StyleBART is more flexible and training-efficient compared with baselines (Jin et al., 2020; Zhan et al., 2022) which require training models for all style and task combinations.

Specifically, StyleBART utilizes style adapters as plug-and-play modules for the style control. The style adapters are learned at the pretraining stage through the *inverse paraphrasing* task on the style corpus, while the base HG model is learned at the fine-tuning stage on news-headline pairs. During inference, we achieve unsupervised SHG by switching to the target style adapter. In this way, Style-BART has the same high decoding efficiency as Jin et al. (2020), while overcoming its problems in task scheduling and inefficient training.

We evaluate StyleBART using the same three SHG tasks described in Jin et al. (2020). Both automatic and human evaluations demonstrate the superiority of our method over baselines.

## 2 Approach

### 2.1 Problem Formulation

We denote the parallel news-headline dataset as $D = \{\langle x, y \rangle\}$, where each pair $\langle x, y \rangle$ consists of a news article $x$ and its corresponding plain headline $y$. The corpus for the $i^{\text{th}}$ style $s_i$ is denoted as $T^{s_i} = \{t^{s_i}\}$. Our goal is to generate stylistic headline $y^{s_i}$ with style $s_i$ given the news article $x$. Note that this is an unsupervised setup, as no news article and stylistic headline pairs are available.

### 2.2 StyleBART Architecture

As shown in Figure 2, our model consists of two parts: the BART model and the adapter modules. We use the BART as the base model for StyleBART. Adapters (Rebuffi et al., 2018) are light-weight bottleneck layers inserted into a base model. It is designed as a parameter-efficient method to fine-

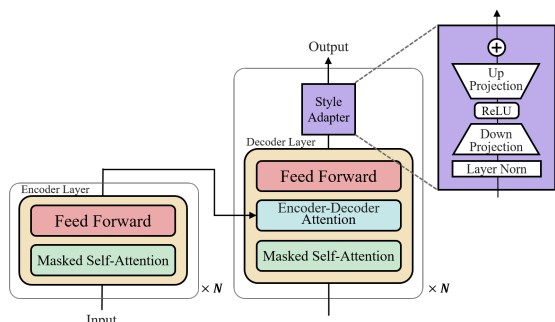

Figure 2: The architecture of StyleBART consists of the BART model and the adapter modules. The pretrained BART model is composed of $N$ encoder layers and $N$ decoder layers. The adapter module is inserted after the feed-forward sublayer in each decoder layer.

tune the base model for a new task (Houlsby et al., 2019), language (Pfeiffer et al., 2020) or domain (Bapna et al., 2019). In StyleBART, we instead use the adapters to control the style of the model output.

Formally, following Houlsby et al. (2019), an adapter module $A$ is composed of layer normalization (LN) of the input $z \in R^h$, down-projection ($W_{\text{down}} \in R^{h \times b}$) with the bottleneck dimension $b$, non-linear function (ReLU), up-projection ($W_{\text{up}} \in R^{b \times h}$), and a residual connection with the input $z$:

$$A(z) = W_{\text{up}}^T \text{ReLU}(W_{\text{down}}^T \text{LN}(z)) + z. \quad (1)$$

We insert the adapters into each decoder layer of the base model. For each style, we have a set of style adapters. In the following, we use $\theta^b$ to denote the set of parameters of the BART model, and $\theta^{s_i}$ to denote that of the style adapters for style $s_i$.

### 2.3 Training and Inference

We divide the training process of StyleBART into three steps (Figure 3): (1) Style adapter pretraining, which learns the style adapters to control the style of the model output by pretraining on the style dataset $T^{s_i}$; (2) Headline generation fine-tuning, which optimizes the base model to generate plain headline on the headline dataset $D$; (3) Stylistic headline generation, which generates stylistic headline by switching to the target style adapters during inference.

**Step 1: Style Adapter Pretraining.** The style adapters can be trained with the style dataset $T^{s_i}$ by the denoising auto-encoding task, which reconstructs the sentence $t^{s_i}$ from $g_n(t^{s_i})$. Here $g_n$ is a noise function that generates a perturbed version

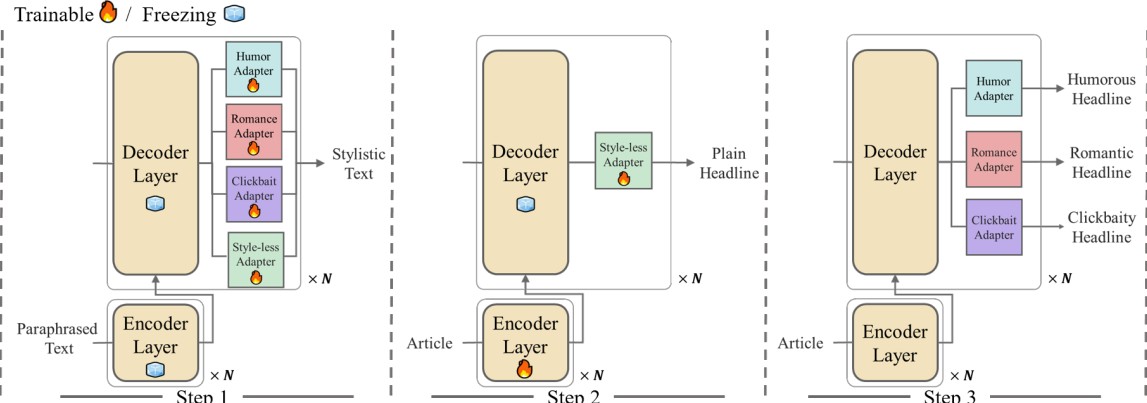

Figure 3: Training and inference framework of StyleBART. Step 1: Style adapter pretraining; Step 2: Headline generation fine-tuning; Step 3: Stylistic headline generation.

of the input, such as token masking and token deletion used during BART pretraining (Lewis et al., 2020) and our baselines (Jin et al., 2020; Zhan et al., 2022). However, this training method is suboptimal.

Following Krishna et al. (2020), the *style* can be loosely defined as the common patterns of lexical choice and syntactic constructions that are distinct from the content of a sentence. Considering how the noise function $g_n$ works, we argue that the corrupted text $g_n(t^{s_i})$ still contains the stylistic information. As a result, the model may learn to guide the style of its output with the style of its input, which deviates from our goal to control the output style with style adapters. We call this undesirable *spurious correlation* (Gu et al., 2019). Instead, we propose to address this issue with *inverse paraphrasing* method.

Inspired by Krishna et al. (2020), we replace $g_n(t^{s_i})$ with $g_p(t^{s_i})$, which is generated by feeding $t^{s_i}$ to a paraphrase model[1] trained to maximize diversity. Table 1 displays examples of stylistic sentences and their perturbed versions with the noise function $g_n$ and the paraphrase model $g_p$. Intuitively, the paraphrase model can better strip away information from $t^{s_i}$ that is predictive of its original style. As a result, our model has to rely on the style adapters instead of the input sentence to control its output style, enhancing the adapters' ability for style control.

Specifically, to train the adapters for style $s_i$, we first feed each sentence $t^{s_i}$ to the paraphrase model and get the perturbed sentence $g_p(t^{s_i})$. Then we train the corresponding style adapters while freezing all other parameters with inverse paraphrasing:

$$\hat{\theta}^{s_i} = \arg\max_{\theta^{s_i}} \sum_{t^{s_i} \in T^{s_i}} \log p(t^{s_i} \mid g_p(t^{s_i}); \theta^{s_i}, \theta^b).$$
(2)

Since the input is style-agnostic and the other parameters are freezing, the model has to rely on the style adapters to control its output style. We also learn a style-less adapter $\theta^{s_0}$ in the same way except replacing $T^{s_i}$ with a dataset $T^{s_0}$, which consists of plain text from BART pretraining.[2]

**Step 2: Headline Generation Fine-tuning.** In this step, StyleBART learns headline generation on the headline dataset $D = \{\langle x, y \rangle\}$. Since the headline is style-less, we insert the style-less adapters into StyleBART and finetune it on the headline generation task. This step is required to force the model to learn the task of headline generation. During finetuning, we only update the parameters of the encoder while freezing the style-less adapters and all other model parameters:

$$\hat{\theta}^b_{enc} = \arg\max_{\theta^b_{enc}} \sum_{<x,y> \in D} \log p(y \mid x; \hat{\theta}^{s_0}, \theta^b).$$
(3)

In this way, we limit the computational cost, and more importantly, mitigate the catastrophic forgetting problem in style control, thus facilitating the switching of different style adapters in Step 3.

**Step 3: Stylistic Headline Generating.** To generate a headline in a given style, we can achieve this by replacing the style-less adapters of StyleBART with the corresponding style-specific adapters.

---

[1]We use the STRAP model (Krishna et al., 2020).

[2]We choose plain text from BART pretraining instead of the headline generation dataset in order to separate the style and downstream task at the data level.

| Style $s_i$ | Stylistic Sentence $t^{s_i}$ | Noised Text $g_n(t^{s_i})$ | Paraphrased Text $g_p(t^{s_i})$ |
|---|---|---|---|
| Humor | There are few things more awkward on a blind date than looking up from your phone to realize she's left. She obviously wasn't blind at all. | She [MASK] blind at all. There are few things [MASK] on a blind date than [MASK] looking up from your phone to realize she's left. | The fact that she left, and the fact that she's obviously not blind, is much more awkward. |
| Romance | He confessed to her that she was his most precious jewel, and that she would always be the rarest gem in his eyes. | [MASK] to her that she was his most [MASK] jewel, and that she would [MASK] rarest gem in his eyes. | He confessed that she was the most valuable thing in his eyes, and he would never see another one. |
| Clickbait | Unbelievable meeting Bob Dylan in Europe for the third time twenty years. | Unbelievable meeting Bob Dylan in [MASK] for the third time twenty years. | The extraordinary meeting with Bob Dylan in Europe for three times. |

Table 1: Examples of sentences and their perturbed versions in three styles: humor, romance and clickbait. The noised text indicates the output after the noise function $g_n$, and the paraphrased text indicates the output after passing through the paraphrase model $g_p$.

StyleBART has already learned the task of headline generation. Therefore, by switching the style-less adapters to the style-specific adapters, we can obtain a style-specific headline generation model. We use $\hat{\theta}^b$, $\hat{\theta}^{s_0}$ and $\hat{\theta}^{s_i}$ to denote the model parameters of the base model, the style-less adapters and the style adapters after Step 2. Then given a news article, StyleBART outputs its style-specific headline $y^{s_i}$ with

$$y^{s_i} = \arg\max_y p(y \mid x; \hat{\theta}^{s_i}, \hat{\theta}^b). \qquad (4)$$

## 3 Experiments

### 3.1 Datasets

Following Jin et al. (2020), the experiment datasets consist of a headline generation dataset CNN-NYT, and three stylistic text datasets for humorous, romantic and clickbaity. The CNN-NYT dataset consists of 146K news-headline pairs from two sources: the New York Times (Sandhaus, 2008) and the CNN dataset (Hermann et al., 2015). We use the same dataset split as Jin et al. (2020). Each style dataset contains 500K style-specific sentences. The humor and romance datasets are collected from the novels in the corresponding genres of BookCorpus (Zhu et al., 2015). The clickbait dataset is obtained from The Examiner-SpamClickBait News dataset.[3] To train the style-less adapters, we also build a style-less dataset, which consists of 500K sentences randomly sampled from BookCorpus,

one of the datasets used in BART pretraining. For style datasets, we use the same split as Jin et al. (2020). For the style-less dataset, we randomly sample 3,000 sentences for both the validation and test set, leaving the rest as the training set.

### 3.2 Baselines

We compare StyleBART with the following baselines:
- **Neural Headline Generation (NHG)**: Finetuning all parameters of BART on the plain headline generation dataset.
- **Two-Step Decoding(TSD)**: A two-step decoding method that first generates a plain headline and then injects style with an unsupervised style transfer model (Dai et al., 2019).
- **Multitask**: A multitask framework that jointly trains on plain headline generation and stylistic text denoising with the BART model.
- **TitleStylist (Jin et al., 2020)**: An approach similar to Multitask, but with carefully designed parameter sharing and switching between tasks.
- **S-SHG (Zhan et al., 2022)**[4]: An approach that first constructs the stem and syntax of the headline similarly to TitleStylist, and then populates that with substantive context.

### 3.3 Evaluation Metrics

**Automatic Evaluation.** We use automatic evaluation to measure the quality and style of the gener-

---

[3]https://www.kaggle.com/datasets/therohk/examine-the-examiner

[4]Our reimplemented S-SHG performs slightly worse than the original paper. However, the paper provides no open-source code or implementation details.

| Style | Method | Generation Quality | | | | | Style Strength |
| | | R1(↑) | R2(↑) | RL(↑) | BERT(↑) | PPL(↓) | PPL-S(↓) |
|---|---|---|---|---|---|---|---|
| Style-less | NHG | 29.7 | 10.9 | 26.3 | 88.2 | 52.6 | - |
| | StyleBART-N | 29.4 | 10.7 | 26.2 | 88.1 | 55.6 | - |
| Humor | TSD* | 20.3 | 5.5 | 18.5 | - | - | - |
| | MultiTask | **30.0** | **11.1** | **26.6** | **87.9** | 52.8 | 1267.6 |
| | TitleStylist | 27.7 | 10.0 | 24.5 | 87.5 | **39.4** | 640.9 |
| | S-SHG | 26.2 | 8.2 | 22.5 | 87.4 | 107.4 | 702.2 |
| | StyleBART | 28.2 | 9.9 | 25.0 | 87.6 | 42.5 | **602.9** |
| Romance | TSD* | 20.5 | 5.8 | 18.7 | - | - | - |
| | MultiTask | **30.0** | **11.4** | **26.8** | **88.1** | 53.2 | 1543.0 |
| | TitleStylist | 27.6 | 9.9 | 24.5 | 87.5 | 39.5 | 740.1 |
| | S-SHG | 26.2 | 8.6 | 22.5 | 87.4 | 101.0 | 685.0 |
| | StyleBART | 27.4 | 9.3 | 24.3 | 87.4 | **36.7** | **560.5** |
| Clickbait | TSD* | 20.3 | 7.1 | 23.2 | - | - | - |
| | MultiTask | **30.3** | **11.3** | **27.0** | **88.3** | 52.2 | 392.4 |
| | TitleStylist | 28.2 | 10.1 | 25.2 | 87.8 | **39.1** | 256.4 |
| | S-SHG | 27.0 | 8.8 | 23.5 | 87.6 | 67.9 | 234.3 |
| | StyleBART | 25.3 | 7.7 | 23.0 | 87.5 | 50.9 | **115.5** |

Table 2: Automatic evaluation results of ROUGE, BERTScore, PPL and PPL-S. TSD* result is directly cited from Li et al. (2022), while all other methods are implemented by ourselves or with the open-sourced code. StyleBART-N: StyleBART with the style-less adapters.

ated headline. For the quality, we use ROUGE (Lin, 2004) and BERTScore (Zhang* et al., 2020) to measure the relevance of the generated headlines to the reference headline and PPL to evaluate its language fluency. The PPL is calculated with OpenAI GPT2 (Radford et al., 2019) finetuned on the plain headlines following Zhan et al. (2022). To evaluate the style, we use PPL-S, which computes the PPL score of the generated headline using GPT2 finetuned on the corresponding style corpus.

**Human Evaluation.** We conducted a human evaluation to more comprehensively assess our model. We randomly sampled 50 news articles from each test set and asked three judges to rate the outputs from TitleStylist, S-SHG and our Style-BART.[5] To assess generation quality, we ask the judges to rate from 1 to 10 (integer values) from three aspects: 1) Relevance—how semantically relevant the headline is to the news article. 2) Attractiveness—how appealing they feel the headline is. 3) Fluency—how comprehensive and easy-to-read the headline is. We then report the average score for each aspect across all test samples and judges. For the style evaluation, we ask the judges to choose the best headline of each style from an entire set of TitleStylist, S-SHG and StyleBART's outputs. We then report the percentage of the times

each model is chosen as the best.

### 3.4 Model Configuration

We implemented StyleBART with the pretrained BART model using Hugging Face. We set the bottleneck dimension $b$ of the adapters to be $64$. To pretrain the adapters, we use the AdamW optimizer (Kingma and Ba, 2014) with $\beta_1 = 0.9$ and $\beta_2 = 0.999$. The learning rate is $5e - 5$. For fine-tuning on the headline generation dataset, we use the same AdamW optimizer and learning rate. For all steps of training, we use a batch size of 8. At inference time, we use beam search with a beam size of 4. All experiments are run on a single NVIDIA RTX 2080ti GPU, except that LLaMA2-InsTuning is run on a single NVIDIA A100 40G GPU

### 4 Results and Discussion

#### 4.1 Automatic Evaluation Results

Table 2 shows the automatic evaluation results of our proposed StyleBART and all baselines.

**Content Relevance.** We use ROUGE (R1,R2, and RL) and BERTScore (BERT) to measure the relevance of the generated headlines to the reference headline. As can be seen, NHG, StyleBART-N, and MultiTask achieve the highest relevance score. This can be explained by that more stylistic headlines would lose some relevance as 1) the reference headlines are style-less; 2) the stylistic head-

---

[5]We do not include TSD, Multitask as they are much worse according to previous work and our automatic evaluation.

| Style | Method | Relevance | Attraction | Fluency |
|---|---|---|---|---|
| Humor | TitleStylist | 5.82 | 5.60 | 7.40 |
| | S-SHG | 6.13 | 7.12 | 8.23 |
| | StyleBART | **7.04** | **7.58** | **8.42** |
| Romance | TitleStylist | 6.12 | 6.06 | 7.78 |
| | S-SHG | 6.37 | 6.04 | 7.63 |
| | StyleBART | **6.70** | **7.30** | **8.40** |
| Clickbait | TitleStylist | 6.00 | 6.16 | 8.00 |
| | S-SHG | 6.17 | 6.65 | 8.17 |
| | StyleBART | **6.76** | **6.96** | **8.26** |

Table 3: Human evaluation results on three aspects.

| Style | TitleStylist | S-SHG | StyleBART |
|---|---|---|---|
| Humor | 20% | 34% | **46%** |
| Romance | 22% | 28% | **50%** |
| Clickbait | 32% | 26% | **42%** |

Table 4: The percentage of choices (%) for the most humorous, romantic or clickbaity headlines among TitleStylist, S-SHG, and StyleBART.

line may use more words outside the news body for improved creativity (Jin et al., 2020). For stylistic headline generation, TSD has the worst relevance score. StyleBART performs slightly worse than TitleStylist (-0.6 averaged RL and -0.1 averaged BERT) and better than S-SHG (+1.3 averaged RL and +0.0 averaged BERT), validating StyleBART's ability in this aspect.

**Language Perplexity.** We use PPL on the GPT2 fine-tuned on plain headlines to measure the fluency of the generated headline. As can be seen, StyleBART surpasses all baselines by a significant margin except that it is slightly worse than TitleStylist. This may be because the headlines from StyleBART have stronger style and more stylistic headlines would also lose some PPL.

**Style Strength.** We use PPL-S to measure the style strength of the generated headline. As can be seen, StyleBART generates headlines with the strongest style across all stylistic headline generation tasks. Compared with the baseline TitleStylist (resp. S-SHG), StyleBART obtains 119.5 (resp. 114.2) lower averaged PPL-S score. The baseline MultiTask performs the worst in style control. Moreover, StyleBART disentangles style learning and headline generation learning, thus only trains once on the headline generation dataset to support all three styles. In contrast, all baselines except TSD require training three times on the headline generation dataset for the three stylistic generation tasks.[6]

### 4.2 Human Evaluation Results

Table 3 and Table 4 present the human evaluation results.

**Generation Quality.** We assess generation quality in relevance, attraction, and fluency, as shown in Table 3. StyleBART performs the best in all three aspects compared to baselines. The results are slightly inconsistent with automatic quality evaluation. This may be explained by that automatic quality evaluation metrics favor less stylistic headlines, while humans do not have such bias.

**Style Strength.** We measure the style strength in Table 4. As can be seen, StyleBART has the highest average selection percentage, followed by the S-SHG model. This is consistent with what we find in the automatic measurement.

### 4.3 Comparison with LLMs

In this part, we compare StyleBART with methods using Large Language Models (LLMs), as presented in Table 5. For GPT3.5-prompting, we perform few-shot prompting with the gpt-3.5-turbo[7] API. We provide stylistic sentences and news-plain headline pairs in the prompt and query the model to generate stylistic headlines for the input news. For LLaMA2-InsTuning, we conduct instruction tuning with LoRA on LLaMA2 (Touvron et al., 2023) for both the inverse paraphrasing task and the news headline generation task. Then we perform stylistic headline generation during testing. Appendix A provides more details. We find that StyleBART overall generates headlines with the best content relevance and the strongest style, demonstrating the superiority of StyleBART even in the era of large language models.[8]

### 4.4 Ablation Study

**Design Choices of Style Adapter Pretraining.** Table 6 shows the effect of different design choices at the style adapter pretraining step. StyleBART−para represents training the adapters

---

[6]The baselines can also support multiple styles within one model by jointly training on a headline generation dataset and all style corpora. However, the performance will decrease according to Jin et al. (2020).

[7]https://platform.openai.com/docs/models/gpt-3-5
[8]Note that StyleBART and these two LLM-based methods are not strictly comparable due to the difference in model size and training data. We include the comparison as a reference.

| Style | Method | Generation Quality | | | | | Style Strength |
|-------|--------|------|------|------|--------|--------|--------|
| | | R1(↑) | R2(↑) | RL(↑) | BERT(↑) | PPL(↓) | PPL-S(↓) |
| Humor | GPT3.5-prompting | 22.7 | 5.9 | 19.5 | 85.8 | 1972.4 | 1161.3 |
| | LLaMA2-InsTuning | 24.5 | 7.7 | 21.6 | 86.3 | **41.7** | 622.8 |
| | StyleBART | **28.2** | **9.9** | **25.0** | **87.6** | 42.5 | **602.9** |
| Romance | GPT3.5-prompting | 22.4 | 5.8 | 19.4 | 85.9 | 1561.2 | 1329.4 |
| | LLaMA2-InsTuning | 26.6 | 8.5 | 23.5 | 87.0 | 112.6 | 879.4 |
| | StyleBART | **27.4** | **9.3** | **24.3** | **87.4** | **36.7** | **560.5** |
| Clickbait | GPT3.5-prompting | 24.5 | 6.9 | 21.0 | 86.2 | 1074.0 | 4094.6 |
| | LLaMA2-InsTuning | **27.2** | **8.9** | **23.8** | 86.9 | **46.2** | 353.1 |
| | StyleBART | 25.3 | 7.7 | 23.0 | **87.5** | 50.9 | **115.5** |

Table 5: Comparison results with methods based on large language models.

| Style | Method | RL | BERT | PPL | PPL-S |
|-------|--------|----|------|-----|-------|
| Style-less | StyleBART-N | 26.2 | 88.1 | 55.6 | - |
| | −para | 26.4 | 88.1 | 57.4 | - |
| | −$s_0$ adapters | 26.5 | 88.2 | 57.6 | - |
| Humor | StyleBART | 25.0 | 87.6 | **42.5** | **602.9** |
| | −para | **26.0** | **88.0** | 54.4 | 1033.0 |
| | −$s_0$ adapters | 18.5 | 86.2 | 221.0 | 1620.0 |
| Romance | StyleBART | 24.3 | 87.4 | **36.7** | **560.5** |
| | −para | **26.2** | **88.1** | 51.8 | 1275.6 |
| | −$s_0$ adapters | 18.9 | 86.4 | 209.6 | 1824.9 |
| Clickbait | StyleBART | 23.0 | 87.5 | 50.9 | **115.5** |
| | −para | **24.2** | **87.7** | **43.3** | 125.8 |
| | −$s_0$ adapters | 21.0 | 87.1 | 53.6 | 134.2 |

Table 6: Ablation on different design choices for style adapter pretraining (Step 1). −para: StyleBART with adapter pretraining by the denoising task. −$s_0$ adapters: StyleBART without pretraining the style-less adapters.

| Style | Method | | | RL | BERT | PPL | PPL-S |
|-------|-----|------|-----|----|------|-----|-------|
| | enc | catt | dec | | | | |
| Style-less | ✓ | × | × | 26.9 | 88.2 | 55.6 | - |
| | ✓ | ✓ | × | 26.7 | 88.2 | 57.8 | - |
| | ✓ | ✓ | ✓ | 26.4 | 88.2 | 55.1 | - |
| Humor | ✓ | × | × | 25.0 | 87.6 | **42.5** | **602.9** |
| | ✓ | ✓ | × | 26.1 | 88.0 | 47.2 | 860.8 |
| | ✓ | ✓ | ✓ | **26.2** | **88.2** | 46.6 | 1011.3 |
| Romance | ✓ | × | × | 24.3 | 87.4 | **36.7** | **560.5** |
| | ✓ | ✓ | × | 25.6 | 88.0 | 45.2 | 911.7 |
| | ✓ | ✓ | ✓ | **26.2** | **88.2** | 47.2 | 1172.4 |
| Clickbait | ✓ | × | × | 23.0 | 87.5 | 50.9 | **115.5** |
| | ✓ | ✓ | × | 23.4 | 87.6 | 50.3 | 139.5 |
| | ✓ | ✓ | ✓ | **24.8** | **87.9** | **42.6** | 156.5 |

Table 7: Ablation on different choices of trainable parameters for headline generation fine-tuning (Step2). enc: the encoder part. catt: the cross attention part. dec: the decoder part.

with the denoising task instead of the inverse paraphrasing task. StyleBART−$s_0$ adapters fine-tune the BART model without style-less adapters at step 2, thus without pretraining the style-less adapters. As can be seen, all methods perform similarly in style-less headline generation. When it comes to stylistic headline generation, StyleBART−$s_0$ adapters decrease dramatically in both the summarization quality and the style control. The inverse paraphrasing task is crucial for style control, while slightly decreasing the relevance of the generated headline to the reference headline. This may be because the reference headlines are style-less, while the headlines produced by using the inverse paraphrasing task have a stronger style.

**Trainable Parameters of Headline Generation Fine-tuning.** We compare different choices of trainable parameters at the headline generation fine-tuning step, as illustrated in Table 7. As can be seen, all fine-tuning methods get similar scores

when generating style-less headlines, indicating that fine-tuning only partial parameters is enough for learning the headline generation task. When comparing on the SHG task, only updating the encoder can best control the style of the generated headlines among all fine-tuning methods. As the style adapters are inserted at the decoder, freezing the decoder and cross-attention can better maintain their style control ability obtained during pretraining.

### 4.5 Case Study

In this section, we present examples of generated stylistic headlines, as shown in Table 8. Again, we concentrate on StyleBART, TitleStylist, and S-SHG, as they outperform the other methods.

From this example as well as others, we find that all three methods generate relevant and fluent headlines. However, StyleBART is better at style control. For example, the phrase "won't last forever"

| | | |
|---|---|---|
| News Abstract | | Cyber monday has been the biggest single shopping day of the year for online retailers. But retailers are spreading their online sales throughout the thanksgiving holiday. As a result, cyber monday's growth is flattening. Analyst: cyber monday will phase out eventually. |
| Humor | TitleStylist | Cyber monday: the biggest shopping day of the year? |
| | S-SHG | Cyber monday is the biggest single shopping day of the year for online retailers |
| | StyleBART | Cyber monday flattens out for retailers |
| Romance | TitleStylist | Cyber monday: the biggest single shopping day of the year? |
| | S-SHG | Cyber monday is still the biggest single shopping day of the year for online retailers |
| | StyleBART | Cyber monday won't last forever |
| Clickbait | TitleStylist | Cyber monday is flattening |
| | S-SHG | Cyber monday has become the biggest single shopping day of the year for online retailers |
| | StyleBART | Is cyber monday the end of online shopping? |

Table 8: Examples of the stylistic headlines generated by different methods.

used in the romantic headline by StyleBART is a common expression in romantic contexts. Style-BART uses questions to raise the reader's curiosity in clickbaity headline. We also observe that when generating headlines of different styles, StyleBART produces more diverse results, while TitleStylist and S-SHG are more likely to change only a few words or expressions.

### 4.6 Extension to Stylistic Story Telling

To test whether StyleBART can flexibly combine tasks and styles, we conduct experiments on stylistic story telling. Specifically, giving the first sentence of a story, this task generates story follow-ups with a desired style. We use the ROCStories dataset (Mostafazadeh et al., 2016) as the standard story telling dataset which contains around 100,000 stories. We randomly select 4080 stories for both validation and test set and leave the rest as the training set. With the dataset, we fine-tune our base model following Step 2 (Section 2.3) and combine the fine-tuned base model with existing humor/romance/clickbait style adapters for inference. In this way, we efficiently build a stylistic story telling model which supports three styles while only requiring fine-tuning once.

Table 9 shows the automatic evaluation results. As can be seen, when switching to the style adapters, the model generates more stylistic stories (PPL-S), while the generated stories become less relevant to the reference story follow-ups (BERT). This is consistent with stylistic headline generation, demonstrating that our style adapters can be com-

| Style | Method | BERT | PPL | PPL-S |
|---|---|---|---|---|
| Style-less | NST | 88.5 | 4.4 | - |
| | StyleBART-N | 88.5 | 4.7 | - |
| Humor | | -0.5 | +0.4 | -17.0 |
| Romance | StyleBART | -0.4 | +0.4 | -14.4 |
| Clickbait | | -1.6 | +23.1 | -968.7 |

Table 9: Automatic evaluation results of stylistic story telling. NST: fine-tune all BART parameters on the plain story telling data. For StyleBART, we report the score changes with respect to the StyleBART-N model.

bined with different downstream tasks to control the style of the generated text.

## 5 Related Work

### 5.1 Headline Generation

Headline generation is the task of generating relevant and concise headlines for given news. It has various application scenarios, such as automated news writing (Li et al., 2022) and product advertising (Kanungo et al., 2022).

Traditional approaches on headline generation rely on linguistic features and handcrafted rules (Dorr et al., 2003; Knight and Marcu, 2002). With the advancement of neural networks, neural headline generation shows its capacity to generate high quality headlines (Rush et al., 2015). However, controlling the style of these headlines remains challenging. Jin et al. (2020) propose the first unsupervised stylistic headline generation model which relies only a standard headline generation dataset

and mono-style corpora. They design a multitask learning framework to jointly learn both plain headline generation and stylistic text denoising with carefully designed parameter sharing and switching strategy. Zhan et al. (2022) further extend by decomposing the headline into style and content. They define stem and syntax as the style and generate the style first using a similar multitask framework as Jin et al. (2020). After that, substantive context is populated into that style with the conditional masked language modelling task. However, these works jointly learn style and headline generation, thus cannot support freely combination of styles and generation tasks at inference time.

## 5.2 Unsupervised Text Style Transfer

Text style transfer is to generate a sentence consistent with desired style while preserving the content of the source sentence (Shen et al., 2017). Due to the scarcity of style parallel corpora, unsupervised text style transfer is widely explored in previous works. Wu et al. (2019) regard text style transfer as a cloze task to accomplish sentiment style conversion of sentences. Dai et al. (2019) incorporates the sentiment style information through a reconstruction task. Krishna et al. (2020) redefine the style transfer problem as a paraphrase generation problem and propose a method based on reverse paraphrasing to generate the desired style. Lai et al. (2021) utilize large pretrained models such as BART and GPT-2, using BLEU scores and style classification scores as rewards, to achieve significant improvements in the transformation of formal and informal text. These models can be combined with plain headline generation model to achieve stylistic headline generation. However, they require two-steps decoding at inference time, while StyleBART generates stylistic headlines in one decoding step.

## 6 Conclusion

We propose StyleBART, an unsupervised stylistic generation method, which enables to freely combine the downstream tasks and styles by disentangling style learning and downstream task learning. Experimental results show that our model can generate content-relevant and style-intensive headlines, and can be extended to other stylistic generation tasks.

## Limitations

This work mainly explores the stylistic headline generation task. We leave the exploration of more combinations of tasks and styles such as stylistic machine translation, stylistic document summarization as future work. At the same time, our current method achieves stylistic generation by simply switching the adapters, which cannot provide fine-grained control of the style. This hinders Style-BART from meeting the diverse user demands related to style control.

## Ethics Statement

We present a training-efficient approach to build an unsupervised stylistic headline generation model which disentangles the headline generation learning and style learning. Despite the strong performance of style control, StyleBART inherits the societal impacts including some negative ones of the original BART model, such as societal biases (Milios and BehnamGhader, 2022) and misuse of language models (Tamkin et al., 2021). The implicit biases are expected to be removed by debiasing either the dataset or the model (Meade et al., 2022; Zhou et al., 2022). StyleBART makes it possible to generate text in various (e.g. clickbait) style which can be used to propagate the malicious or offensive content (Welbl et al., 2021). Future explorations are needed to mitigate the misuse of StyleBART model.

## Acknowledgements

This project was supported by National Natural Science Foundation of China (No. 62106138, No. 62306132) and Shanghai Sailing Program (No. 21YF1412100). We thank the anonymous reviewers for their insightful feedbacks on this work.

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

| | Task: News Style Headline Generation |
|---|---|

**Task: News Style Headline Generation**

**Stage 1: Learning Text Styles**
Please read the following representative text with a particular style and understand its stylistic features.
Style Text: [STYLE TEXT]
· · ·

**Stage 2: Learning Headline Generation**
Here are some examples of news and corresponding headlines to learn how to generate news headlines:
News: [NEWS]
Headline: [HEADLINE]
· · ·

**Stage 3: Generating Stylized Headlines**
Generate a news headline with the previously learned style based on the news text given above:
News: [NEWS]
Style headline:

Table 10: The data template used for GPT3.5-prompting.

| Stage | | Template |
|---|---|---|
| Training | | **Inverse Paraphrasing Task** |
| | Instruction | Convert the following sentence to its [STYLE] version. |
| | Input | Text: [PARAPHRASED TEXT] |
| | Output | [STYLE] version text: [STYLISTIC TEXT] |
| | | **Headline Generation Task** |
| | Instruction | Read the following news abstract and write a headline for it. |
| | Input | News: [NEWS] |
| | Output | Headline: [HEADLINE] |
| Inference | | **Stylistic Headline Generation** |
| | Instruction | Read the following news abstract and a write a [STYLE] version headline of the news. |
| | Input | News: [NEWS] |
| | Output | [STYLE] version headline: |

Table 11: The data template used for LLaMA2-InsTuning training and inference stage.

# A   Appendix

## A.1   GPT3.5-prompting Details

We perform few-shot prompting with the gpt-3.5-turbo API. We follow the unsupervised setup of StyleBART which assumes paired news-stylistic headlines are unavailable. As a results, we use three non-parallel stylistic sentences, five news-plain headline pairs in our data template. Table 10 shows the details. We set the the temperature to 0.5.

## A.2   LLaMA2-InsTuning Details

We sample 10,000 data points per task[9] and construct the corresponding instruction data. Table 11 shows our data template in training and inference stages. We utilize the 40,000 instruction data jointly to fine-tune LLaMA2-7B with LoRA. The hyperparameters for the LoRA module are set as follows: $lora\_rank = 256$, $lora\_alpha = 256$, and $lora\_dropout = 0.05$. The training process employs the AdamW optimizer with parameters $\beta_1 = 0.9, \beta_2 = 0.999$. We adopt a learning rate of $2e - 5$. The training is executed with a batch size of 80 for one epoch, amounting to a total of 500 steps. After our training stage, we directly use the inference template in Table 11 to evaluate the fine-tuned model.

[9]Our training loss shows no significant reductions after 150 steps when using a batch size of 80. Therefore, we do not use the full dataset as larger dataset requires more computation and only brings marginal performance improvement.

## A.3   Comparison with TitleStylist-BART

TitleStylist is initialized using the MASS model, a pretrained model with an encoder-decoder structure similar to BART. We adopt the methodology employed in TitleStylist and apply it to the BART model. We execute both TitleStylist-MASS (with their open-source code) and TitleStylist-BART (with our reimplemented code). As shown in Table 12, We can find that these two variations yield similar ROUGE and BERTScore results while TitleStylist-MASS gets better PPL and PPL-S scores. Therefore, we present the results for TitleStylist-MASS in this paper, as it demonstrates better performance and is consistent with the original paper.

| Style | Method | Generation Quality | | | | | Style Strength |
| | | R1(↑) | R2(↑) | RL(↑) | BERT(↑) | PPL(↓) | PPL-S(↓) |
|---|---|---|---|---|---|---|---|
| Humor | TitleStylist-BART | 28.2 | 10.0 | 24.9 | 87.7 | 68.5 | 839.3 |
| | TitleStylist-MASS | 27.7 | 10.0 | 24.5 | 87.5 | 39.4 | 640.9 |
| Romance | TitleStylist-BART | 28.4 | 10.1 | 25.0 | 87.7 | 62.6 | 923.7 |
| | TitleStylist-MASS | 27.6 | 9.9 | 24.5 | 87.5 | 39.5 | 740.1 |
| Clickbait | TitleStylist-BART | 28.2 | 9.8 | 24.9 | 87.7 | 48.2 | 192.9 |
| | TitleStylist-MASS | 28.2 | 10.1 | 25.2 | 87.8 | 39.1 | 256.4 |

Table 12: Comparison results of TitleStylist-BART and Titleylist-MASS.