# OpenReview forum: "StyleBART: Decorate Pretrained Model with Style Adapters for Unsupervised Stylistic Headline Generation"
_EMNLP/2023/Conference — EMNLP 2023 Findings_

### Official Review · Reviewer_vmEE · 2023-07-27

**Soundness:** 4

**Excitement:**

3: Ambivalent: It has merits (e.g., it reports state-of-the-art results, the idea is nice), but there are key weaknesses (e.g., it describes incremental work), and it can significantly benefit from another round of revision. However, I won't object to accepting it if my co-reviewers champion it.

**Paper Topic And Main Contributions:**

Stylized headline generation is the task of generating headlines in a particular style (e.g. humorous, clickbait, etc). This paper presents a new approach for stylized headline generation using style adapters. Adapters are small modules that are added to a pretrained model, and are trained to perform a particular task while keeping the rest of the model parameters fixed. This paper proposes to use one adapter to learn the task of headline generation, and another for the task of style transfer using inverse paraphrasing, a technique introduced in [1]. The entire approach operates in an unsupervised setting, and does not require any paired (article, stylized headline) data.

The authors conduct a thorough set of experiments on the task of stylized headline generation, and show that their approach is more effective than existing methods in terms of both automatic metrics and human evaluation. The authors also conduct several ablation studies, qualitative analysis and experiments on stylized story generation.

[1] - https://arxiv.org/abs/2010.05700

**Reasons To Accept:**

1. This paper studies a practically relevant problem in controlled text generation - stylized headline generation.

2. The paper proposes a simple new approach for stylized headline generation, using paraphrasing and style adapters. Human evaluations show that this approach is more effective than existing methods in terms of style transfer accuracy, headline quality and fluency.

3. This paper has a thorough set of experiments, and includes both automatic and human evaluation. The paper also has a good set of ablation studies showcasing the beneift of inverse paraphrasing and style adapters. Finally, the authors also add a qualitative analysis of the generated headlines. The authors also study the benefit of their adapters on stylised story generation, showcasing the modularity of their approach.

**Reasons To Reject:**

1. This paper would be more relevant to the current paradigms with experiments on modern large language models. This includes both open-source models like LLAMA [2] which can be fine-tuned, and closed-source models with just API access like ChatGPT / GPT3.5. The paper currently only does experiments on BART, which was introduced in 2019. It will be interesting to see how much better the proposed approach is after fine-tuning adapters or LoRA [3] on LLAMA, and how that compares to few-shot prompting on ChatGPT / GPT3.5.

2. *Minor*: it would be easier to compare between systems in Table 2,3,4 if a single aggregated score is reported, by combining the metrics using the sentence-level aggregation method in [1].

3. *Minor*: the method introduced in the paper is quite similar to [1]. The main differences are --- 1) the use of style adapters instead of separate models; 2) evaluation on stylized headline generation and storytelling instead of style transfer.

**Overall**: The paper is well written and has a good experimental setup. The proposed approach, while similar to [1], is effective at headline generation and much more modular than [1] due to the use of style adapters. However, weakness #1 is a major concern for me, which is what reduced my "excitement score". But I am open to further discussion and will raise the excitement score if the authors will add similar experiments on LLAMA in the camera ready version, or add some few-shot experiments with large language models as a baseline.

[1] - https://arxiv.org/abs/2010.05700
[2] - https://arxiv.org/abs/2302.13971
[3] - https://arxiv.org/abs/2106.09685

**Reproducibility:**

4: Could mostly reproduce the results, but there may be some variation because of sample variance or minor variations in their interpretation of the protocol or method.

**Reviewer Confidence:**

4: Quite sure. I tried to check the important points carefully. It's unlikely, though conceivable, that I missed something that should affect my ratings.

**Typos Grammar Style And Presentation Improvements:**

L006: "researches" --> "research"
L044: "requires" --> "require"

---

> ### Author Rebuttal · Authors · 2023-08-29
>
> **Q1. This paper would be more relevant to the current paradigms with experiments on modern large language models. This includes both open-source models like LLAMA [2] which can be fine-tuned, and closed-source models with just API access like ChatGPT / GPT3.5. The paper currently only does experiments on BART, which was introduced in 2019. It will be interesting to see how much better the proposed approach is after fine-tuning adapters or LoRA [3] on LLAMA, and how that compares to few-shot prompting on ChatGPT / GPT3.5. **
>
> R1. Thank you for your suggestion. We conduct comparison experiments with few-shot prompting on GPT3.5 and instruction tuning on LLAMA2.
>
> (1) Few-shot prompting on GPT3.5. We perform few-shot prompting with the gpt-3.5-turbo API. We provide five stylistic sentences and three news-headline pairs in the prompt and query the model to generate stylistic headlines for the input news. The results show that our method generates headlines with not only more relevant content (+4.1 average RL and +1.5 average BERTScore), but also stronger style (-1768.8 average PPL-S ). We will include this in the revised version.
>
> (2) Instruction tuning on LLAMA2. We conduct instruction tuning using LoRA on both the inversing paraphrasing task and the news headline generation task. Then we perform stylistic headline generation at test time. We sample 10,000 data points from each training dataset to train, as larger dataset requires more computation and only brings marginal performance improvement (Our training loss curve shows no significant reductions after 150 steps when using a batch size of 80.) The results show that StyleBART generates headlines with more relevant content (+1.8 average RL and +0.8 average BERTScore) and stronger style(-125.8 average PPL-S). We will add this in the revised version.
>
> We do not conduct experiments that directly apply our framework to LLAMA2 for two reasons:
>
> (1) It is a common practice to use instructions to fine-tune the LLM like LLAMA2, instead of tuning with non-instruction data (this is what we do in our proposed framework).
>
> (2) BART is an encoder-decoder model while LLAMA2 is decoder only. As a result, we cannot directly apply StyleBART to LLAMA2 (for example, at step 2 we only finetune the encoder of BART but LLAMA2 has no encoder).
>
> As a result, we conduct experiments on LLAMA2 with instruction tuning. Due to the limited computation resource and time, we will add the experiment results that directly apply StyleBART to LLAMA2 in the revised version.
>
> | Style     | Method           | R1   | R2   | RL   | BERT | PPL    | PPL-S  |
> | --------- | ---------------- | ---- | ---- | ---- | ---- | ------ | ------ |
> | Humor     | GPT3.5-prompting | 22.7 | 5.9  | 19.5 | 85.8 | 1972.4 | 1161.3 |
> | Humor     | LLaMA-InsTuning  | 24.1 | 7.5  | 21.5 | 86.6 | 38.9   | 536.2  |
> | Humor     | ours             | 28.2 | 10.0 | 24.9 | 87.6 | 42.5   | 602.9  |
> | Romance   | GPT3.5-prompting | 22.4 | 5.8  | 19.4 | 85.9 | 1561.2 | 1329.4 |
> | Romance   | LLaMA-InsTuning  | 24.3 | 7.9  | 21.6 | 86.5 | 39.1   | 825.7  |
> | Romance   | ours             | 27.4 | 9.4  | 24.3 | 87.4 | 36.7   | 560.5  |
> | Clickbait | GPT3.5-prompting | 24.5 | 6.9  | 21.0 | 86.2 | 1074.0 | 4094.6 |
> | Clickbait | LLaMA-InsTuning  | 27.1 | 9.1  | 23.6 | 87.1 | 41.5   | 294.4  |
> | Clickbait | ours             | 25.3 | 7.7  | 23.0 | 87.5 | 50.9   | 115.5  |
>
> **Q2.** ***Minor*: it would be easier to compare between systems in Table 2,3,4 if a single aggregated score is reported, by combining the metrics using the sentence-level aggregation method in [1].**
>
> R2. Thank you for your suggestions. We will add the sentence-level aggregation method in [1] in a revised version.
>
> **Q3.** ***Minor*: the method introduced in the paper is quite similar to [1]. The main differences are --- 1) the use of style adapters instead of separate models; 2) evaluation on stylized headline generation and storytelling instead of style transfer.**
>
> R3. Thank you for your question. We agree that StyleBART is similar to [1] in learning the style with the inverse paraphrasing task. However, [1] only focuses on style transfer, while our focus is style control in specific tasks like headline generation/storytelling. This makes our method different from [1] in three aspects.
>
> (1) Our problem is more complex than [1]. While [1] only learns on the mono-style text dataset,  we learn on both the task (headline generation/storytelling) dataset and the mono-style text dataset. This makes our problem more complex as we need to carefully design model architecture and training framework to transfer learning between these two datasets.
>
> (2) The model proposed in [1] cannot directly applied in our setup except using the two-step-decoding method which first generates a plain headline and then injects style with [1]. However, the two-step-decoding method brings additional decoding cost and may suffer from the error propagation problem.
>
> (3) One of the key contributions of our method is to provide a modular solution for unsupervised style control in specific text generation tasks, which supports task and style combination at inference time. However, [1] fails to provide such solutions as it only focuses on unsupervised style transfer.

---

### Official Review · Reviewer_B3qY · 2023-08-05

**Soundness:** 3

**Excitement:**

3: Ambivalent: It has merits (e.g., it reports state-of-the-art results, the idea is nice), but there are key weaknesses (e.g., it describes incremental work), and it can significantly benefit from another round of revision. However, I won't object to accepting it if my co-reviewers champion it.

**Paper Topic And Main Contributions:**

This paper proposed an unsupervised stylistic headline generation model which contains three steps. It separates training the style and the generation modules, then combines them as needed. The experimental results are based on the CNN-NYT and three stylistic text datasets. The proposed StyleBART achieves SOTA performance in the unsupervised stylistic headline generation task.

**Reasons To Accept:**

1. A novel scenario for stylistic headline generation. It is easy to implement and can achieve better performance.
2. The paper is well-organized and easy to follow.
3. This paper showcases thorough experiments on multiple datasets and various backbones.

**Reasons To Reject:**

1. The setups for the task, and the contributions are limited. The idea is not novel and the paper is not more engaging.
2. There have been no tests to determine the statistical significance of the results.

**Reproducibility:**

4: Could mostly reproduce the results, but there may be some variation because of sample variance or minor variations in their interpretation of the protocol or method.

**Reviewer Confidence:**

4: Quite sure. I tried to check the important points carefully. It's unlikely, though conceivable, that I missed something that should affect my ratings.

---

> ### Author Rebuttal · Authors · 2023-08-29
>
> **Q1. The setups for the task, and the contributions are limited. The idea is not novel and the paper is not more engaging.**
>
> R1. Thank you for the comments. Our contributions are :
>
> (1) State-of-the-art performance: we propose a simple and effective approach for the unsupervised stylistic headline generation task, and achieve SOTA performance in both automatic and human evaluation.
>
> (2) A more modular approach: our proposed method StyleBART is much modular than previous work, which supports flexible combinations of downstream tasks and styles at inference time.
>
> (3) Better style learning method: we propose inverse paraphrasing to learn the style, which works better than the previous denoising auto-encoding task.
>
> To make the paper more engaging, we compare StyleBART with methods using LLM. We find that StyleBART works better. We will add this in the revised version.
>
> (1) Few-shot prompting on GPT3.5. We perform few-shot prompting with the gpt-3.5-turbo API. We provide five stylistic sentences and three news-headline pairs in the prompt and query the model to generate stylistic headlines for the input news. The experimental results show that our approach generates headlines with more relevant content (+4.1 average RL and +1.5 average BERTScore) and stronger style (-1768.8 average PPL-S).
>
> (2) Instruction tuning on LLAMA2. We conduct instruction tuning with LoRA on both the inversing paraphrasing task and the news headline generation task. Then we perform stylistic headline generation at test time. We sample 10,000 data points from each training dataset to train, as larger dataset requires more computation and only brings marginal performance improvement (Our training loss curve shows no significant reductions after 150 steps when using a batch size of 80.) The results show that StyleBART generates headlines with more relevant content (+1.8 average RL and +0.8 average BERTScore) and stronger style (-125.8 average PPL-S).
>
> | Style     | Method           | R1   | R2   | RL   | BERT | PPL    | PPL-S  |
> | --------- | ---------------- | ---- | ---- | ---- | ---- | ------ | ------ |
> | Humor     | GPT3.5-prompting | 22.7 | 5.9  | 19.5 | 85.8 | 1972.4 | 1161.3 |
> | Humor     | LLaMA-InsTuning  | 24.1 | 7.5  | 21.5 | 86.6 | 38.9   | 536.2  |
> | Humor     | ours             | 28.2 | 10.0 | 24.9 | 87.6 | 42.5   | 602.9  |
> | Romance   | GPT3.5-prompting | 22.4 | 5.8  | 19.4 | 85.9 | 1561.2 | 1329.4 |
> | Romance   | LLaMA-InsTuning  | 24.3 | 7.9  | 21.6 | 86.5 | 39.1   | 825.7  |
> | Romance   | ours             | 27.4 | 9.4  | 24.3 | 87.4 | 36.7   | 560.5  |
> | Clickbait | GPT3.5-prompting | 24.5 | 6.9  | 21.0 | 86.2 | 1074.0 | 4094.6 |
> | Clickbait | LLaMA-InsTuning  | 27.1 | 9.1  | 23.6 | 87.1 | 41.5   | 294.4  |
> | Clickbait | ours             | 25.3 | 7.7  | 23.0 | 87.5 | 50.9   | 115.5  |
>
> **Q2. There have been no tests to determine the statistical significance of the results.**
>
> R2. Thanks for the suggestion. We will add this in a revised version.

---

### Official Review · Reviewer_ZP6W · 2023-08-06

**Soundness:** 3

**Excitement:**

3: Ambivalent: It has merits (e.g., it reports state-of-the-art results, the idea is nice), but there are key weaknesses (e.g., it describes incremental work), and it can significantly benefit from another round of revision. However, I won't object to accepting it if my co-reviewers champion it.

**Paper Topic And Main Contributions:**

This work proposes StyleBART, an unsupervised approach for stylistic headline generation. This method decorates the pretrained BART model with adapters that are responsible for different styles and allows the generation of headlines with diverse styles by simply switching the adapters. This work separates the task of style learning and headline generation, making it possible to freely combine the base model and the style adapters during inference.

**Questions For The Authors:**

1. The TitleStylist paper used a encoder-decoder model and train it from scratch. In your experiments, do you use the method from TitleStylist and apply it to BART model as the baseline? If not, then some results are not fair.

**Reasons To Accept:**

1. This work shows good style strength as shown in Table 2 and enables one model for multiple styles with simply multiple sets of style adapters.
2. The ablation study of different sets of parameters for headline generation training as shown in Table 6 is interesting and intriguing.
3. The proposed paraphrasing method for learning the style from corpus is interesting.

**Reasons To Reject:**

1. As shown in Table 2 and 6, fine-tuning only the encoder of BART in step 2 can promote the style strength but it can also hurt the headline generation quality (RL score tells the difference).
2. The baseline TitleStylist has proposed finetuning some layer normalization and cross-attention parameters in the decoder to learn the style, while this work proposes using adapter. Adapter has been introduced in 2020 and has been shown to be good at switching among different settings. So the innovation here is quite incremental.
3. In this era of LLMs, I am wondering whether few-shot in context learning of a LLM like Vicuna-13B can accomplish this task very well.


**Reproducibility:**

4: Could mostly reproduce the results, but there may be some variation because of sample variance or minor variations in their interpretation of the protocol or method.

**Reviewer Confidence:**

5: Positive that my evaluation is correct. I read the paper very carefully and I am very familiar with related work.

---

> ### Author Rebuttal · Authors · 2023-08-29
>
> **Q1. As shown in Table 2 and 6, fine-tuning only the encoder of BART in step 2 can promote the style strength but it can also hurt the headline generation quality (RL score tells the difference).**
>
> R1. Thank you for the question.
>
> (1) According to Table 6 and line 381-385, finetuning only the encoder gets similar scores with other finetuning methods when generating style-less headlines, indicating that finetuning only the encoder is sufficient for learning on the headline generation task.
>
> (2) We agree that finetuning only the encoder gets lower RL score and BERTScore when generating stylistic headlines, as shown in Table 6. However, this does not necessarily mean that finetuning only the encoder hurts the headline generation quality.  Finetuning only the encoder generates more stylistic headlines (lower PPL-S score in Table 6). As is stated in line 308-312, more stylistic headlines would lose some RL score and BERTScore as 1) the reference headlines are style-less; 2) the stylistic headline may use more words outside the news body for improved style strength.
>
>
>
> **Q2. The baseline TitleStylist has proposed finetuning some layer normalization and cross-attention parameters in the decoder to learn the style, while this work proposes using adapter. Adapter has been introduced in 2020 and has been shown to be good at switching among different settings. So the innovation here is quite incremental.**
>
> R2. We agree that TitleStylist finetunes some layer normalization and cross-attention parameters in the decoder to learn the style, while StyleBART uses adapters to learn the style. However, our method is not simply switching the style control module.
>
> (1) We propose a modular solution for unsupervised stylistic text generation.  As pointed out by reviewer vmEE, our method is much more modular than TitleStylist. As shown in Figure 1, StyleBART can flexibly combine the downstream tasks and the styles at inference time, while TitleStylist needs retraining for each task and style combination. This is achieved through the proposed model architecture and pretraining-finetuning framework of StyleBART instead of just using adapters.
>
> (2) We propose the inverse paraphrasing task to learn the style adapters, while TitleStylist learns the style on the denoising auto-encoding task. As demonstrated in Table 5, the inverse paraphrasing task is important to strengthen the adapters' style control ability.
>
> **Q3. In this era of LLMs, I am wondering whether few-shot in context learning of a LLM like Vicuna-13B can accomplish this task very well.**
>
> R3. Thank you for your suggestion. We conduct few-shot experiments using GPT3.5 (gpt-3.5-turbo). We provide five stylistic sentences and three news-headline pairs in the prompt and query the model to generate stylistic headlines for the input news.  The experimental results show that our approach generates headlines with more relevant content (+4.1 average RL and +1.5 average BERTScore) and a stronger style (-1768.8 average PPL-S). We will add this in the revised version.
>
> | Style     | Method           | R1   | R2   | RL   | BERT | PPL    | PPL-S  |
> | --------- | ---------------- | ---- | ---- | ---- | ---- | ------ | ------ |
> | Humor     | GPT3.5-prompting | 22.7 | 5.9  | 19.5 | 85.8 | 1972.4 | 1161.3 |
> | Humor     | ours             | 28.2 | 10.0 | 24.9 | 87.6 | 42.5   | 602.9  |
> | Romance   | GPT3.5-prompting | 22.4 | 5.8  | 19.4 | 85.9 | 1561.2 | 1329.4 |
> | Romance   | ours             | 27.4 | 9.4  | 24.3 | 87.4 | 36.7   | 560.5  |
> | Clickbait | GPT3.5-prompting | 24.5 | 6.9  | 21.0 | 86.2 | 1074.0 | 4094.6 |
> | Clickbait | ours             | 25.3 | 7.7  | 23.0 | 87.5 | 50.9   | 115.5  |
>
> **Q4. The TitleStylist paper used a encoder-decoder model and train it from scratch. In your experiments, do you use the method from TitleStylist and apply it to BART model as the baseline? If not, then some results are not fair.**
>
> R4. Thank you for pointing out this.
>
> (1) Instead of training from scratch, TitleStylist is initialized from the MASS model, which is a pre-trained model similar to BART with an encoder-decoder structure.
>
> (2) We run both TitleStylist-MASS (with their open-sourced code) and TitleStylist-BART (with our reimplemented code). We find that they get similar ROUGE and BERTScore, while TitleStylist-MASS gets better PPL and PPL-S scores. Therefore, we report TitleStylist-MASS in our paper, which works better and is consistent with the original paper. We will add TitleStylist-BART in the revised version.
>
> | **Style** | **Method**        | **R1** | **R2** | **RL** | **BERT** | **PPL** | **PPL-S** |
> | --------- | ----------------- | ------ | ------ | ------ | -------- | ------- | --------- |
> | Humor     | TitleStylist-BART | 28.2   | 10.0   | 24.9   | 87.7     | 68.5    | 839.3     |
> | Humor     | TitleStylist-MASS | 27.7   | 10.0   | 24.5   | 87.5     | 39.4    | 640.9     |
> | Romance   | TitleStylist-BART | 28.4   | 10.1   | 25.0   | 87.7     | 62.6    | 923.7     |
> | Romance   | TitleStylist-MASS | 27.6   | 9.9    | 24.5   | 87.5     | 39.5    | 740.1     |
> | Clickbait | TitleStylist-BART | 28.2   | 9.8    | 24.9   | 87.7     | 48.2    | 192.9     |
> | Clickbait | TitleStylist-MASS | 28.2   | 10.1   | 25.2   | 87.8     | 39.1    | 256.4     |

---

### Meta-Review · Area_Chair_tPVr · 2023-09-19

**Recommendation:** 3

**Metareview:**

This paper proposes an unsupervised stylistic headline generation model.

The method is easy to implement and can achieve good performance.
The paper is well-organized and easy to follow.
The experiments are reasonably thorough, including both automatic and human evaluation.

Several reviewers comment that the novelty is limited and similar methods have been proposed before.
Inclusion of statistical significance would strengthen the results.
Also, lack of comparison to LLMs is hindering its relevance among current research directions.

---

### Decision · Program_Chairs · 2023-10-07

**Decision:**

Accept-Findings

**Comment:**

This paper proposes an unsupervised stylistic headline generation model.

The method is easy to implement and can achieve good performance.
The paper is well-organized and easy to follow.
The experiments are reasonably thorough, including both automatic and human evaluation.

Several reviewers comment that the novelty is limited and similar methods have been proposed before.
Inclusion of statistical significance would strengthen the results.
Also, lack of comparison to LLMs is hindering its relevance among current research directions.